# Scattering Characteristics of a Circularly Polarized Bessel Pincer Light-Sheet Beam Interacting with a Chiral Sphere of Arbitrary Size

**DOI:** 10.3390/mi16080845

**Published:** 2025-07-24

**Authors:** Shu Zhang, Shiguo Chen, Qun Wei, Renxian Li, Bing Wei, Ningning Song

**Affiliations:** 1School of Physics, Xidian University, Xi’an 710071, China; qunwei@xidian.edu.cn (Q.W.); rxli@mail.xidian.edu.cn (R.L.); bwei@xidian.edu.cn (B.W.); nnsong@stu.xidian.edu.cn (N.S.); 2Key Laboratory of Optoelectronic Information Perception in Complex Environment, Ministry of Education, Xi’an 710071, China

**Keywords:** circular polarization, Bessel pincer light-sheet beam, chiral particle, VSWFs, BSCs, GLMT

## Abstract

The scattering interaction between a circularly polarized Bessel pincer light-sheet beam and a chiral particle is investigated within the framework of generalized Lorenz–Mie theory (GLMT). The incident electric field distribution is rigorously derived via the vector angular spectrum decomposition method (VASDM), with subsequent determination of the beam-shape coefficients (BSCs) $p_{mn}^{u}$ and $q_{mn}^{u}$ through multipole expansion in the basis of vector spherical wave functions (VSWFs). The expansion coefficients for the scattered field ($A_{mn}^{s}$$B_{mn}^{s}$) and interior field ($A_{mn}$$B_{mn}$) are derived by imposing boundary conditions. Simulations highlight notable variations in the scattering field, near-surface field distribution, and far-field intensity, strongly influenced by the dimensionless size parameter ka, chirality κ, and beam parameters (beam order *l* and beam scaling parameter $\alpha_{0}$). These findings provide insights into the role of chirality in modulating scattering asymmetry and localization effects. The results are particularly relevant for applications in optical manipulation and super-resolution imaging in single-molecule microbiology.

## 1. Introduction

Chirality, an intrinsic asymmetry where an object cannot be superimposed on its mirror image [1], profoundly impacts nature and technology, particularly in medicine and biology, where enantiomers exhibit drastically different pharmacological effects. Chiral materials, with applications spanning pharmaceutical analysis [2], food safety [3], agriculture [4], and environmental protection [5], enable stereoselective sensing and enantioselective catalysis, with recent advances in chiral electromagnetic waves enhancing responses via metamaterials [6,7,8,9,10,11] for improved biosensing [12,13], separation [14,15], catalysis [16], and imaging [17,18]. Critically, chiral particles are also vital for optical manipulation, where circular polarization-dependent chiral forces offer novel methods for sensing, separation, and enantioselective reactions; consequently, optically manipulating chiral particles is a key research focus with transformative potential. However, effective optical manipulation fundamentally relies on understanding light–matter interactions, quantified by scattering, extinction, and absorption cross-sections; precise control of chiral particles—essential for advancing applications—is difficult without insights into their scattering behavior. Therefore, studying the scattering properties of chiral particles is not only necessary but foundational for unlocking their potential in optical manipulation and related fields.

Researchers have employed a combination of numerical and analytical methods to gain a deeper understanding of the interaction between electromagnetic waves and chiral particles. Numerical methods, such as T-matrix [19,20], Method of Moments (MoM) [21,22,23,24], Finite-Difference Time-Domain (FDTD) [25], and Finite-Difference Frequency-Domain (FDFD) [26], are widely used for studying complex geometries and materials. These methods, although powerful, require substantial computational resources and are limited by accumulated errors and model-size constraints. On the other hand, analytical solutions offer faster, more stable, and highly accurate results, making them invaluable for verifying numerical solutions and performing rapid calculations. Although typically limited to simpler geometries, analytical methods are crucial for understanding fundamental scattering mechanisms. Key contributions to this field include Schneider’s [27] initial overview of scattering theory for chiral particles and Gordon’s [28] extension of classical Mie theory to account for circular dichroism and optical rotatory distribution. Bohren [29,30], Hinder [31], Cooray [32], and more recently, Demir et al. [25] have further advanced the field by providing exact solutions and practical tools for analyzing scattering by chiral structures. Despite this progress, the scattering characteristics of newly developed structured light beams, such as Bessel pincer beams [33], are not yet well understood.

Laser technology advancements have sparked interest in structured light beams, particularly regarding their interactions with particles for applications such as biomedical imaging [34], material processing [35], and quantum information [36]. Among the diverse structured beams under investigation, Gaussian [37], Hermite–Gaussian [38], and Bessel beams [39,40,41,42] have stood out due to their distinctive propagation characteristics. Several studies [43,44,45,46,47,48,49] have investigated the scattering of Gaussian beams, highlighting limitations such as shape recovery and diffraction. In contrast, Bessel beams have demonstrated distinct advantages, encompassing diffraction-free propagation and self-healing [50,51,52,53,54,55,56,57], which are particularly beneficial in nonlinear optics, particle acceleration, and optical metrology. These attributes have stimulated further research on Bessel beams, such as Qu et al.’s [58] analysis of their interactions with chiral dielectric spheres and Li et al.’s [59,60] exploration of their utilization in particle trapping and separation. Compared with traditional Bessel beams, linearly polarized Bessel pincer light sheets [61,62] offer enhanced autofocusing and bending capabilities, promising even greater flexibility and precision in particle manipulation, and marking them as promising tools for advanced applications [63,64,65]. However, detailed studies on their scattering interactions with chiral particles are still lacking. Notably, the interaction of a chiral particle with a circularly polarized Bessel pincer light-sheet beam, which combines the unique propagation dynamics of pincer beams with the intrinsic chirality of circular polarization, remains unexplored. Investigating this specific configuration is crucial, as it promises deeper insights into chirality-dependent optical forces and torques, potentially unlocking new avenues for highly selective manipulation and characterization of chiral matter in the aforementioned applications.

This study aims to investigate the scattering characteristics of a circularly polarized Bessel pincer light sheet interacting with a chiral sphere of arbitrary size. This research addresses the existing gap in understanding these interactions and explores their potential applications in optical manipulation [66] and sensing. Section 2 presents the theoretical framework, deriving beam-shape coefficients ($p_{mn}^{u}$, $q_{mn}^{u}$) under circular polarization, which serves as a foundation for comprehending light–sphere interactions [67]. Utilizing Condon–Rosenfeld relations [68] and spherical vector wave functions (SVWFs) [69], we derive series expansions for incident, internal, and scattered fields to provide a comprehensive description of scattering behavior. Section 3 focuses on numerical simulations and analysis, including an in-depth examination of scattered-field characteristics as well as near-field and far-field scattering phenomena. This analysis aims to elucidate how beam parameters and particle properties influence scattering outcomes. Finally, Section 4 summarizes key findings, emphasizing how this study enhances our understanding of structured light’s interactions with chiral particles. While this work focuses on the fundamental scattering behavior, the derived field expansion coefficients $A_{mn}^{s}$ and $B_{mn}^{s}$ provide critical inputs for predicting optical forces/torques via the Maxwell stress tensor [70,71,72]. This establishes a direct pathway toward particle manipulation schemes, which will be implemented in future experimental studies.

## 2. Methods

Now, consider the expansion coefficients of the incident Bessel pincer light-sheet beam. Assuming a circularly polarized Bessel pincer light-sheet beam (located in (*y*, $z_{0}$)) in the free space, with a time dependence varying as exp(−iωt) (but removed from the field equations), a left-circularly polarized (LCP) or right-circularly polarized (RCP) vector potential function A is considered, as shown in Figure 1. In the present analysis, the evanescent waves are not considered. The vector angular spectrum function is expressed [33,61,62] as(1)$$A^{u}=\int_{-\infty }^{+\infty }S_{l}\left(p,q\right)\cdot e^{ik\left[py+q\left(z-z_{0}\right)\right]}dp\left({\hat{e}}_{y}\pm i{\hat{e}}_{x}\right)$$
where(2)$$S_{l}\left(p,q\right)=\frac{kE_{0}}{\pi \alpha_{0}^{l}\sqrt{\alpha_{0}^{2}-p^{2}}}ℜ\left\{{\left(p+i\sqrt{\alpha_{0}^{2}-p^{2}}\right)}^{l}\right\}$$

### 2.1. Angle of Bessel Pincer Light-Sheet Beam

In the above expression, $E_{0}$ is the electric field amplitude. *k* is the wave number of the incident beam. ℜ is the real part of a complex number, $\alpha_{0}$ is the beam scaling parameter of the Bessel pincer light sheet, and *l* is the order of the spherical Bessel functions. In the angular spectrum decomposition method, each individual plane-wave component propagates along the direction (p,q) in the transverse plane, (p,q) are the directional cosines (p=sinα, q=cosα), and α is the angle of propagation of the individual plane wave. $e_{x}$ and $e_{y}$ are the unitary vectors. As shown in Equation (Equation 1), the LCP (or RCP) potential is formed as a combination of two dephased linearly polarized Bessel pincer light sheets. The + sign denotes the LCP case, whereas the − sign represents the RCP state. The superscript *u* indicates the type of polarization (lc or rc).

Using Lorenz’s gauge condition [72,73], the electric field vector is expressed as(3)$$E^{u}=ik\left[A^{u}+\nabla \left(\nabla \cdot A^{u}\right)/k^{2}\right]$$

The substitution of Equation (Equation 1) into Equation (Equation 3) and algebraic manipulation leads to the expressions for the Cartesian components of the incident wave field as(4)$$E_{x}^{u}=\mp k\int_{-\infty }^{+\infty }S_{l}\cdot e^{ik\left[py+q\left(z-z_{0}\right)\right]}dp$$(5)$$E_{y}^{u}=ik\int_{-\infty }^{+\infty }S_{l}\cdot e^{ik\left[py+q\left(z-z_{0}\right)\right]}\left(1-p^{2}\right)dp$$(6)$$E_{z}^{u}=-ik\int_{-\infty }^{+\infty }S_{l}\cdot e^{ik\left[py+q\left(z-z_{0}\right)\right]}pqdp$$

Equations (Equation 4)–(Equation 6) are the Cartesian components of the incident electric field vector, with LCP and RCP waves. In contrast with the previous results for linearly polarized Bessel pincer light sheets, for which the axial electric field vector component is zero, Equation (Equation 4) shows that $E_{x}^{u}$ is finite and changes sign when the polarization is switched from LCP to RCP. Moreover, the expressions of $E_{y}^{u}$ and $E_{z}^{u}$ remain unaltered as the polarization is changed from LCP to RCP.

Correspondingly, the magnetic field vector under the Lorenz gauge is expressed as(7)$$H^{u}={\left(\varepsilon \mu \right)}^{1/2}\nabla \times A^{u}$$

First, consider the scenario where Bessel pincer light sheets are left-handed circularly polarized upon incidence. When u corresponds to left-handed circular polarization (LCP), the expression for the electric field vector can be written as(8)$$E^{lc}\left(y,z\right)=-k\int_{-\infty }^{+\infty }\left[{\hat{e}}_{x}+i\left(p^{2}-1\right){\hat{e}}_{y}+ip\sqrt{1-p^{2}}{\hat{e}}_{z}\right]S_{l}e^{ik\cdot r}e^{-ikqz_{0}}\cos\alpha d\alpha$$
The polynomial $\left[{\hat{e}}_{x}+i\left(p^{2}-1\right){\hat{e}}_{y}+ip\sqrt{1-p^{2}}{\hat{e}}_{z}\right]\cdot e^{ik\cdot r}$ can be expanded using vector sphere harmonic functions and expressed as(9)$$\left[{\hat{e}}_{x}+i\left(p^{2}-1\right){\hat{e}}_{y}+ip\sqrt{1-p^{2}}{\hat{e}}_{z}\right]\cdot e^{ik\cdot r}=\sum_{n=1}^{\infty }\sum_{m=-n}^{n}D_{mn}\left[p_{mn}^{\prime lc}N_{nm}^{\left(1\right)}+q_{mn}^{\prime lc}M_{nm}^{\left(1\right)}\right]$$
where(10)$$D_{mn}=\frac{(2n+1)(n-m)!}{n(n+1)(n+m)!}$$(11)$$\begin{array}{c} q_{mn}^{\prime lc}=-i^{n+1}e^{-im\beta }\left[\pi_{mn}\left(\cos\alpha \right)e_{\theta }\left(\alpha \right)-i\tau_{mn}\left(\cos\alpha \right)e_{\phi }\left(\alpha \right)\right]\\ \times \left[{\hat{e}}_{x}+i\left(p^{2}-1\right){\hat{e}}_{y}+ip\sqrt{1-p^{2}}{\hat{e}}_{z}\right]\;\;\;\;\; \end{array}$$(12)$$\begin{array}{c} p_{mn}^{\prime lc}=-i^{n+1}e^{-im\beta }\left[\tau_{mn}\left(\cos\alpha \right)e_{\theta }\left(\alpha \right)-i\pi_{mn}\left(\cos\alpha \right)e_{\phi }\left(\alpha \right)\right]\\ \times \left[{\hat{e}}_{x}+i\left(p^{2}-1\right){\hat{e}}_{y}+ip\sqrt{1-p^{2}}{\hat{e}}_{z}\right]\;\;\;\;\; \end{array}$$

Substituting Formulas (Equation 9)–(Equation 12) into Equation (Equation 8), leads to(13)$$E^{lc}\left(r,\theta ,\varphi \right)=-\sum_{n=1}^{\infty }\sum_{m=-n}^{n}iE_{mn}\left[p_{mn}^{lc}M_{nm}^{\left(1\right)}\left(kr\right)+q_{mn}^{lc}N_{nm}^{\left(1\right)}\left(kr\right)\right]$$
and(14)$$q_{mn}^{lc}=k\frac{D_{mn}^{1/2}}{-i^{n+1}}\int_{-\infty }^{\infty }q_{mn}^{\prime lc}S_{l}\left(\alpha \right)e^{-ikqz_{0}}\cos\alpha d\alpha$$(15)$$p_{mn}^{lc}=k\frac{D_{mn}^{1/2}}{-i^{n+1}}\int_{-\infty }^{\infty }p_{mn}^{\prime lc}S_{l}\left(\alpha \right)e^{-ikqz_{0}}\cos\alpha d\alpha$$

Consider the mathematical relationship between the unit vectors in Cartesian coordinates and spherical coordinates:(16)$$\begin{array}{c} {\hat{e}}_{x}=-{\hat{e}}_{\varphi },\\ {\hat{e}}_{y}=\sin\alpha {\hat{e}}_{r}+\cos\alpha {\hat{e}}_{\theta },\\ {\hat{e}}_{z}=\cos\alpha {\hat{e}}_{r}-\sin\alpha {\hat{e}}_{\theta } \end{array}$$

Subsequently, Equations (Equation 11) and (Equation 12) can be rewritten, respectively, as(17)$$q_{mn}^{\prime lc}=i^{n}e^{-im\beta }\left[\tau_{mn}\left(\cos\alpha \right)-q\pi_{mn}\left(\cos\alpha \right)\right]$$(18)$$p_{mn}^{\prime lc}=i^{n}e^{-im\beta }\left[\pi_{mn}\left(\cos\alpha \right)-q\tau_{mn}\left(\cos\alpha \right)\right]$$

Substituting Equation (Equation 17) into Equation (Equation 14), and (Equation 18) into Equation (Equation 15), the incident expansion coefficients for the Bessel pincer light sheets under LCP polarization are expressed as(19)$$q_{mn}^{lc}=ikD_{mn}^{1/2}\int_{-\infty }^{\infty }S_{l}\left(\alpha \right)e^{-ikqz_{0}}e^{-im\beta }\left[\tau_{mn}\left(\cos\alpha \right)-q\pi_{mn}\left(\cos\alpha \right)\right]\cos\alpha d\alpha$$(20)$$p_{mn}^{lc}=ikD_{mn}^{1/2}\int_{-\infty }^{\infty }S_{l}\left(\alpha \right)e^{-ikqz_{0}}e^{-im\beta }\left[\pi_{mn}\left(\cos\alpha \right)-q\tau_{mn}\left(\cos\alpha \right)\right]\cos\alpha d\alpha$$

Analogous to the left-circularly polarized (LCP) scenario, through vector angular spectrum expansion and subsequent mathematical derivation, the expansion coefficients for Bessel pincer light sheets under right-circularly polarized (RCP) incidence are eventually expressed as(21)$$q_{mn}^{rc}=ikD_{mn}^{1/2}e^{-im\beta }\int_{-\infty }^{\infty }\left[\tau_{mn}\left(\cos\alpha \right)+q\pi_{mn}\left(\cos\alpha \right)\right]S_{l}\left(\alpha \right)e^{-ikqz_{0}}\cos\alpha d\alpha$$(22)$$p_{mn}^{rc}=ikD_{mn}^{1/2}e^{-im\beta }\int_{-\infty }^{\infty }\left[\pi_{mn}\left(\cos\alpha \right)+q\tau_{mn}\left(\cos\alpha \right)\right]S_{l}\left(\alpha \right)e^{-ikqz_{0}}\cos\alpha d\alpha$$
where $\pi_{mn}$ and $\tau_{mn}$ are auxiliary functions [69,74], expressed as(23)$$\pi_{mn}\left(\cos\theta \right)=\frac{m}{\sin\theta }P_{n}^{m}\left(\cos\theta \right),\;\;\;\;\;\;\;\;\;\;\;\;\;\;\;\;\;\tau_{mn}\left(\cos\theta \right)=\frac{dP_{n}^{m}\left(\cos\theta \right)}{d\theta }$$

### 2.2. Scattering of a Bessel Pincer Light-Sheet Beam by a Chiral Sphere

Considering a chiral sphere (non-magnetic and nondispersive) located at the origin of the Cartesian coordinate system $O_{xyz}$, it is characterized by its permittivity $\varepsilon_{c}$, permeability $\mu_{c}$, and the chirality parameter κ, with radius *a*. As shown in Figure 2, the chiral sphere is illuminated by a circularly polarized Bessel pincer light sheet propagating along the *z*-axis. The media of a chiral sphere can be described by the following constitutive relations [29,68]:(24)$$\begin{array}{c} D=\varepsilon E+i\zeta_{c}H\\ B=-i\zeta_{c}E+\mu H \end{array}$$
wherein $\zeta_{c}$ denotes the chiral parameter of the medium, which can be real or complex, describing the cross-coupling strength between the electric and magnetic fields with the unit of ΩΩmm. Here, ε is the permittivity and μ is the permeability of the medium. In Equation (Equation 24), the term $i\zeta_{c}H$ in D directly describes the electric displacement component contributed by the magnetic field, while the term $-i\zeta_{c}E$ in B directly describes the magnetic flux density component contributed by the electric field. Notably, the cross-terms originate from the unique electromagnetic coupling effect of chiral media. According to reciprocity, if the magnetic field can induce electric polarization, the electric field must induce magnetic polarization (and vice versa); otherwise, electromagnetic symmetry would be violated. Mathematically, the two cross-terms are characterized by the same parameter $\zeta_{c}$ with opposite signs, satisfying the Lorentz reciprocity theorem. The effect of cross-terms is completely determined by the geometric properties (such as helicity) of the medium itself, without considering spatial distribution (non-local integration). This means that the non-local dependence of D on E in the cross-term $i\zeta_{c}H$, and the non-local dependence of B on H in the cross-term $-i\zeta_{c}E$, are both negligible. Therefore, this effect is local and reciprocal, eliminating the need for complex non-local theories.

By introducing the chirality parameter and using constitutive relations [75,76,77,78] to model isotropic chiral media, Equation (Equation 24) are rewritten as(25)$$\begin{array}{c} D=\varepsilon_{0}\varepsilon_{r}E+i\kappa \sqrt{\mu_{0}\varepsilon_{0}}H\\ B=-i\kappa \sqrt{\mu_{0}\varepsilon_{0}}E+\mu_{r}\mu_{0}H \end{array}$$
wherein $\varepsilon \;=\;\varepsilon_{0}\varepsilon_{r}$ and $\mu =\mu_{r}\mu_{0}$, where $\varepsilon_{r}$ and $\mu_{r}$ denote the relative permittivity and relative permeability of the chiral dielectric sphere, respectively. ε0=8.854×10−12FFmm and μ0=4π×10−7HHmm are the permittivity and permeability of free space, respectively. The chiral parameter is defined as $\zeta_{c}\;=\;\kappa \sqrt{\mu_{0}\varepsilon_{0}}$, where κ is a dimensionless chirality parameter describing the chiral coupling strength of the medium. The parameter κ is governed by the inequality $\kappa^{2}<\varepsilon \mu$, where κ>0 typically corresponds to a “chiral handedness”, κ<0 corresponds to the opposite “chiral handedness”, and κ=0 reduces the relation of the constitutive relation to the constitutive relation of ordinary media.

Based on Borhen’s method [29] and Shang’s previous work [46,76], we provide a brief introduction to beam scattering by a chiral sphere. The internal field of a chiral sphere can be expanded in terms of VSWFs [79] in the following forms:(26)$$E_{\mathrm{int}}=\sum_{n=1}^{+\infty }\sum_{m=-n}^{n}iE_{mn}\left\{\begin{array}{c} A_{mn}M_{mn}^{\left(1\right)}\left(r,k_{R}\right)+A_{mn}N_{mn}^{\left(1\right)}\left(r,k_{R}\right)\\ +B_{mn}M_{mn}^{\left(1\right)}\left(r,k_{L}\right)-B_{mn}N_{mn}^{\left(1\right)}\left(r,k_{L}\right) \end{array}\right\}$$(27)$$H_{\mathrm{int}}=\frac{k_{c}}{i\omega \mu_{c}}\sum_{n=1}^{\infty }\sum_{m=-n}^{m=n}E_{mn}\left\{\begin{array}{c} A_{mn}N_{mn}^{\left(1\right)}\left(r,k_{R}\right)+A_{mn}M_{mn}^{\left(1\right)}\left(r,k_{R}\right)\\ +B_{mn}N_{mn}^{\left(1\right)}\left(r,k_{L}\right)-B_{mn}M_{mn}^{\left(1\right)}\left(r,k_{L}\right) \end{array}\right\}$$
where $A_{mn}$ and $B_{mn}$ represent the unknown expansion coefficients of the internal field [80]. $k_{L}$ and $k_{R}$ are two modes, characterized by the wave numbers $k_{L}=k_{0}\left(\sqrt{\varepsilon \mu }-\kappa \right)$ and $k_{R}=k_{0}\left(\sqrt{\varepsilon \mu }+\kappa \right)$, in such a chiral medium, corresponding to the left-hand and right-hand circular polarizations, respectively. The incident beam and scattered field can be expanded as follows:(28)$$E_{inc}=-\sum_{n=1}^{\infty }\sum_{m=-n}^{n}iE_{mn}\left[p_{mn}^{u}N_{mn}^{\left(1\right)}\left(k_{1}r\right)+q_{mn}^{u}M_{mn}^{\left(1\right)}\left(k_{1}r\right)\right]$$(29)$$H_{inc}=-\frac{k_{1}}{\omega \mu_{1}}\sum_{n=1}^{\infty }\sum_{m=-n}^{n}E_{mn}\left[q_{mn}^{u}N_{mn}^{\left(1\right)}\left(k_{1}r\right)+p_{mn}^{u}M_{mn}^{\left(1\right)}\left(k_{1}r\right)\right]$$(30)$$E_{sca}=\sum_{n=1}^{\infty }\sum_{m=-n}^{n}iE_{mn}\left[A_{mn}^{s}M_{mn}^{\left(3\right)}\left(k_{1}r\right)+B_{mn}^{s}N_{mn}^{\left(3\right)}\left(k_{1}r\right)\right]$$(31)$$H_{sca}=\frac{k_{1}}{i\omega \mu_{1}}\sum_{n=1}^{\infty }\sum_{m=-n}^{n}E_{mn}\left[A_{mn}^{s}N_{mn}^{\left(3\right)}\left(k_{1}r\right)+B_{mn}^{s}M_{mn}^{\left(3\right)}\left(k_{1}r\right)\right]$$
where $E_{mn}=i^{n}\left|E_{0}\right|{\left[\frac{2n+1}{n(n+1)}\frac{(n-m)!}{(n+m)!}\right]}^{1/2}$. $p_{mn}^{u}$ and $q_{mn}^{u}$ are the expansion coefficients of the incident field, while $A_{mn}^{s}$ and $B_{mn}^{s}$ are the unknown scattering coefficients. The wavenumber in the ambient medium is expressed as $k_{1}\;=\;m_{1}k_{0}=\omega \sqrt{\mu_{r,1}\varepsilon_{r,1}}\sqrt{\mu_{0}\varepsilon_{0}}$, where $k_{1}$ denotes the wavenumber in the ambient medium; $m_{1}$ is the refractive index of the ambient medium; $k_{0}=\sqrt{\mu_{0}\varepsilon_{0}}$ is the wavenumber in free space; ω represents the angular frequency; and $\mu_{r,1}$ and $\varepsilon_{r,1}$ are the relative permeability and relative permittivity of the ambient medium, respectively. The vector spherical harmonics $M_{mn}^{\left(j\right)}$ and $N_{mn}^{\left(j\right)}$ are expressed as(32)$$\begin{array}{c} M_{mn}^{\left(j\right)}\left(kr\right)=\left[i\pi_{mn}\left(\cos\theta \right)e_{\theta }-\tau_{mn}\left(\cos\theta \right)e_{\phi }\right]z_{n}^{\left(j\right)}\left(kr\right)exp\left(im\phi \right) \end{array}$$(33)$$\begin{array}{c} N_{mn}^{\left(j\right)}\left(kr\right)l=\left[\tau_{mn}\left(\cos\theta \right)e_{\theta }+i\pi_{mn}\left(\cos\theta \right)e_{\phi }\right]\frac{1}{kr}\frac{d}{dr}\left[rz_{n}^{\left(j\right)}\left(kr\right)\right]exp\left(im\phi \right)\\ \;\;\;\;\;\;\;\;\;\;\;\;\;\;\;\;\;\;\;\;\;\;\;\;\;\;\;\;\;\;\;\;\;\;\;\;\;\;\;\;\;\;\;+e_{r}n\left(n+1\right)P_{n}^{m}\left(\cos\theta \right)\frac{z_{n}^{\left(j\right)}\left(kr\right)}{kr}exp\left(im\phi \right) \end{array}$$
wherein $z_{n}^{\left(j\right)}$ denotes the spherical Bessel function, where *j* = 1, 2, 3 correspond to the first kind, second kind, and third kind of Bessel functions, respectively.

At spherical surface r=a, the electric and magnetic fields inside and outside satisfy the following boundary conditions:(34)$$\begin{array}{c} \hat{r}\times E_{\mathrm{int}}=\hat{r}\times \left(E_{inc}+E_{sca}\right)\\ \hat{r}\times H_{\mathrm{int}}=\hat{r}\times \left(H_{inc}+H_{sca}\right) \end{array}$$
where $\hat{r}$ is the unit normal vector at the spherical surface. Substituting Equations (Equation 26)–(Equation 31) into the boundary conditions (Equation (Equation 34)), via complex algebraic operations, the relationship between $p_{mn}^{u}$, $q_{mn}^{u}$, and $A_{mn}^{s}$, $B_{mn}^{s}$ is as below:(35)$$A_{mn}^{s}=q_{mn}^{u}\cdot \frac{\left(DG-CH\right)}{\left(AH-BG\right)}+p_{mn}^{u}\cdot \frac{\left(FG-EH\right)}{\left(AH-BG\right)}$$(36)$$B_{mn}^{s}=q_{mn}^{u}\cdot \frac{\left(AD-BC\right)}{\left(AH-BG\right)}+p_{mn}^{u}\cdot \frac{\left(AF-BE\right)}{\left(AH-BG\right)}$$
where(37)$$\begin{array}{c} A=\left[\eta_{r}\psi_{n}\left(x_{R}\right){\xi^{\prime }}_{n}\left(x_{1}\right)-\xi_{n}\left(x_{1}\right){\psi^{\prime }}_{n}\left(x_{R}\right)\right]\\ B=\left[\eta_{r}\psi_{n}\left(x_{L}\right){\xi^{\prime }}_{n}\left(x_{1}\right)-\xi_{n}\left(x_{1}\right){\psi^{\prime }}_{n}\left(x_{L}\right)\right]\\ C=\left[\eta_{r}\psi_{n}\left(x_{R}\right){\psi^{\prime }}_{n}\left(x_{1}\right)-\psi_{n}\left(x_{1}\right){\psi^{\prime }}_{n}\left(x_{R}\right)\right]\\ D=\left[\eta_{r}\psi_{n}\left(x_{L}\right){\psi^{\prime }}_{n}\left(x_{1}\right)-\psi_{n}\left(x_{1}\right){\psi^{\prime }}_{n}\left(x_{L}\right)\right]\\ E=\left[\psi_{n}\left(x_{R}\right){\psi^{\prime }}_{n}\left(x_{1}\right)-\eta_{r}\psi_{n}\left(x_{1}\right){\psi^{\prime }}_{n}\left(x_{R}\right)\right]\\ F=\left[\eta_{r}\psi_{n}\left(x_{1}\right){\psi^{\prime }}_{n}\left(x_{L}\right)-{\psi^{\prime }}_{n}\left(x_{1}\right)\psi_{n}\left(x_{L}\right)\right]\\ G=\left[\eta_{r}\xi_{n}\left(x_{1}\right){\psi^{\prime }}_{n}\left(x_{R}\right)-\psi_{n}\left(x_{R}\right){\xi^{\prime }}_{n}\left(x_{1}\right)\right]\\ H=\left[\psi_{n}\left(x_{L}\right){\xi^{\prime }}_{n}\left(x_{1}\right)-\eta_{r}\xi_{n}\left(x_{1}\right){\psi^{\prime }}_{n}\left(x_{L}\right)\right] \end{array}$$

The final expressions for the internal field expansion coefficients $A_{mn}$ and $B_{mn}$ are(38)$$A_{mn}=\eta_{r}\frac{x_{R}}{x_{1}}\left[A_{mn}^{Int\_a}-A_{mn}^{Int\_b}\right]$$(39)$$B_{mn}=\eta_{r}\frac{x_{L}}{x_{1}}\left[B_{mn}^{Int\_a}-B_{mn}^{Int\_b}\right]$$
where(40)$$\begin{array}{c} A_{mn}^{Int\_a}=q_{mn}^{u}\frac{\left[\psi_{n}\left(x_{1}\right){\xi^{\prime }}_{n}\left(x_{1}\right)+{\psi^{\prime }}_{n}\left(x_{1}\right)\xi_{n}\left(x_{1}\right)\right]\left[\eta_{r}{\psi^{\prime }}_{n}\left(x_{L}\right)\xi_{n}\left(x_{1}\right)+\psi_{n}\left(x_{L}\right){\xi^{\prime }}_{n}\left(x_{1}\right)\right]}{\left\{\begin{array}{c} \left[\eta_{r}\cdot \psi_{n}\left(x_{R}\right){\xi^{\prime }}_{n}\left(x_{1}\right)-{\psi^{\prime }}_{n}\left(x_{R}\right)\xi_{n}\left(x_{1}\right)\right]\left[\eta_{r}{\psi^{\prime }}_{n}\left(x_{L}\right)\xi_{n}\left(x_{1}\right)+\psi_{n}\left(x_{L}\right){\xi^{\prime }}_{n}\left(x_{1}\right)\right]\\ -\left[{\psi^{\prime }}_{n}\left(x_{R}\right)\eta_{r}\xi_{n}\left(x_{1}\right)-\psi_{n}\left(x_{R}\right){\xi^{\prime }}_{n}\left(x_{1}\right)\right]\left[{\psi^{\prime }}_{n}\left(x_{L}\right)\xi_{n}\left(x_{1}\right)-\eta_{r}\cdot \psi_{n}\left(x_{L}\right){\xi^{\prime }}_{n}\left(x_{1}\right)\right] \end{array}\right\}}\\ A_{mn}^{Int\_b}=p_{mn}^{u}\frac{\left[\psi_{n}\left(x_{1}\right)\xi_{n}\left(x_{1}\right)+\psi_{n}\left(x_{1}\right){\xi^{\prime }}_{n}\left(x_{1}\right)\right]\left[{\psi^{\prime }}_{n}\left(x_{L}\right)\xi_{n}\left(x_{1}\right)-\eta_{r}\cdot \psi_{n}\left(x_{L}\right){\xi^{\prime }}_{n}\left(x_{1}\right)\right]}{\left\{\begin{array}{c} \left[\eta_{r}\cdot \psi_{n}\left(x_{R}\right){\xi^{\prime }}_{n}\left(x_{1}\right)-{\psi^{\prime }}_{n}\left(x_{R}\right)\xi_{n}\left(x_{1}\right)\right]\left[\eta_{r}{\psi^{\prime }}_{n}\left(x_{L}\right)\xi_{n}\left(x_{1}\right)+\psi_{n}\left(x_{L}\right){\xi^{\prime }}_{n}\left(x_{1}\right)\right]\\ -\left[{\psi^{\prime }}_{n}\left(x_{R}\right)\eta_{r}\xi_{n}\left(x_{1}\right)-\psi_{n}\left(x_{R}\right){\xi^{\prime }}_{n}\left(x_{1}\right)\right]\left[{\psi^{\prime }}_{n}\left(x_{L}\right)\xi_{n}\left(x_{1}\right)-\eta_{r}\cdot \psi_{n}\left(x_{L}\right){\xi^{\prime }}_{n}\left(x_{1}\right)\right] \end{array}\right\}}\\ B_{mn}^{Int\_a}=p_{mn}^{u}\frac{\left[\psi_{n}\left(x_{1}\right)\xi_{n}\left(x_{1}\right)+\psi_{n}\left(x_{1}\right){\xi^{\prime }}_{n}\left(x_{1}\right)\right]\left[\eta_{r}{\psi^{\prime }}_{n}\left(x_{R}\right)\xi_{n}\left(x_{1}\right)-\psi_{n}\left(x_{R}\right){\xi^{\prime }}_{n}\left(x_{1}\right)\right]}{\left\{\begin{array}{c} \left[{\psi^{\prime }}_{n}\left(x_{L}\right)\xi_{n}\left(x_{1}\right)-\eta_{r}\cdot \psi_{n}\left(x_{L}\right){\xi^{\prime }}_{n}\left(x_{1}\right)\right]\left[{\psi^{\prime }}_{n}\left(x_{R}\right)\eta_{r}\xi_{n}\left(x_{1}\right)-\psi_{n}\left(x_{R}\right){\xi^{\prime }}_{n}\left(x_{1}\right)\right]\\ -\left[\eta_{r}{\psi^{\prime }}_{n}\left(x_{L}\right)\xi_{n}\left(x_{1}\right)+\psi_{n}\left(x_{L}\right){\xi^{\prime }}_{n}\left(x_{1}\right)\right]\left[{\psi^{\prime }}_{n}\left(x_{R}\right)\eta_{r}\xi_{n}\left(x_{1}\right)-\psi_{n}\left(x_{R}\right){\xi^{\prime }}_{n}\left(x_{1}\right)\right] \end{array}\right\}}\\ B_{mn}^{Int\_b}=q_{mn}^{u}\frac{\left[\psi_{n}\left(x_{1}\right){\xi^{\prime }}_{n}\left(x_{1}\right)+{\psi^{\prime }}_{n}\left(x_{1}\right)\xi_{n}\left(x_{1}\right)\right]\left[{\psi^{\prime }}_{n}\left(x_{R}\right)\eta_{r}\xi_{n}\left(x_{1}\right)-\psi_{n}\left(x_{R}\right){\xi^{\prime }}_{n}\left(x_{1}\right)\right]}{\left\{\begin{array}{c} \left[{\psi^{\prime }}_{n}\left(x_{L}\right)\xi_{n}\left(x_{1}\right)-\eta_{r}\cdot \psi_{n}\left(x_{L}\right){\xi^{\prime }}_{n}\left(x_{1}\right)\right]\left[{\psi^{\prime }}_{n}\left(x_{R}\right)\eta_{r}\xi_{n}\left(x_{1}\right)-\psi_{n}\left(x_{R}\right){\xi^{\prime }}_{n}\left(x_{1}\right)\right]\\ -\left[\eta_{r}{\psi^{\prime }}_{n}\left(x_{L}\right)\xi_{n}\left(x_{1}\right)+\psi_{n}\left(x_{L}\right){\xi^{\prime }}_{n}\left(x_{1}\right)\right]\left[{\psi^{\prime }}_{n}\left(x_{R}\right)\eta_{r}\xi_{n}\left(x_{1}\right)-\psi_{n}\left(x_{R}\right){\xi^{\prime }}_{n}\left(x_{1}\right)\right] \end{array}\right\}} \end{array}$$

In the expression above, x1=k0a,xR=kRa,xL=kLa,ηr=ε1ε1μ1μ1x1=k0a,xR=kRa,xL=kLa,ηr=ε1ε1μ1μ1εcεcμcμcεcεcμcμc. Considering that the direct calculation of the Riccati–Bessel function and its derivatives tends to diverge when the particle size is large, the logarithmic derivative function $D_{n}^{\left(j\right)}$ of the Riccati–Bessel function is introduced.(41)$$D_{n}^{\left(1\right)}\left(z\right)=\frac{{\psi^{\prime }}_{n}\left(z\right)}{\psi_{n}\left(z\right)},D_{n}^{\left(2\right)}\left(z\right)=\frac{{\chi^{\prime }}_{n}\left(z\right)}{\chi_{n}\left(z\right)},D_{n}^{\left(3\right)}\left(z\right)=\frac{{\xi^{\prime }}_{n}\left(z\right)}{\xi_{n}\left(z\right)}$$

Substituting Formula (Equation 41) into Equations (Equation 35) and (Equation 36), the scattering field expansion coefficients in the form of the logarithmic derivatives of the Ricatti–Bessel functions are rewritten as follows:(42)$$\begin{array}{cc} A_{mn}^{s} & =q_{mn}^{u}\cdot \frac{\psi_{n}\left(x_{1}\right)}{\xi_{n}\left(x_{1}\right)}\frac{\frac{\left[\eta_{r}D_{n}^{\left(1\right)}\left(x_{1}\right)-D_{n}^{\left(1\right)}\left(x_{L}\right)\right]}{\left[D_{n}^{\left(3\right)}\left(x_{1}\right)-\eta_{r}D_{n}^{\left(1\right)}\left(x_{L}\right)\right]}-\frac{\left[\eta_{r}D_{n}^{\left(1\right)}\left(x_{1}\right)-D_{n}^{\left(1\right)}\left(x_{R}\right)\right]}{\left[\eta_{r}D_{n}^{\left(1\right)}\left(x_{R}\right)-D_{n}^{\left(3\right)}\left(x_{1}\right)\right]}}{\frac{\left[\eta_{r}D_{n}^{\left(3\right)}\left(x_{1}\right)-D_{n}^{\left(1\right)}\left(x_{R}\right)\right]}{\left[\eta_{r}D_{n}^{\left(1\right)}\left(x_{R}\right)-D_{n}^{\left(3\right)}\left(x_{1}\right)\right]}-\frac{\left[\eta_{r}D_{n}^{\left(3\right)}\left(x_{1}\right)-D_{n}^{\left(1\right)}\left(x_{L}\right)\right]}{\left[D_{n}^{\left(3\right)}\left(x_{1}\right)-\eta_{r}D_{n}^{\left(1\right)}\left(x_{L}\right)\right]}}\\ \; & +p_{mn}^{u}\cdot \frac{\psi_{n}\left(x_{1}\right)}{\xi_{n}\left(x_{1}\right)}\frac{\frac{\left[\eta_{r}D_{n}^{\left(1\right)}\left(x_{L}\right)-D_{n}^{\left(1\right)}\left(x_{1}\right)\right]}{\left[\eta_{r}D_{n}^{\left(1\right)}\left(x_{L}\right)-D_{n}^{\left(3\right)}\left(x_{1}\right)\right]}-\frac{\left[\eta_{r}D_{n}^{\left(1\right)}\left(x_{R}\right)-D_{n}^{\left(1\right)}\left(x_{1}\right)\right]}{\left[\eta_{r}D_{n}^{\left(1\right)}\left(x_{R}\right)-D_{n}^{\left(3\right)}\left(x_{1}\right)\right]}}{\frac{\left[D_{n}^{\left(1\right)}\left(x_{R}\right)-\eta_{r}D_{n}^{\left(3\right)}\left(x_{1}\right)\right]}{\left[\eta_{r}D_{n}^{\left(1\right)}\left(x_{R}\right)-D_{n}^{\left(3\right)}\left(x_{1}\right)\right]}+\frac{\left[D_{n}^{\left(1\right)}\left(x_{L}\right)-\eta_{r}D_{n}^{\left(3\right)}\left(x_{1}\right)\right]}{\left[\eta_{r}D_{n}^{\left(1\right)}\left(x_{L}\right)-D_{n}^{\left(3\right)}\left(x_{1}\right)\right]}} \end{array}$$(43)$$\begin{array}{cc} B_{mn}^{s} & =q_{mn}^{u}\cdot \frac{\psi_{n}\left(x_{1}\right)}{\xi_{n}\left(x_{1}\right)}\frac{\frac{\left[\eta_{r}D_{n}^{\left(1\right)}\left(x_{1}\right)-D_{n}^{\left(1\right)}\left(x_{L}\right)\right]}{\left[\eta_{r}D_{n}^{\left(3\right)}\left(x_{1}\right)-D_{n}^{\left(1\right)}\left(x_{L}\right)\right]}-\frac{\left[\eta_{r}D_{n}^{\left(1\right)}\left(x_{1}\right)-D_{n}^{\left(1\right)}\left(x_{R}\right)\right]}{\left[\eta_{r}D_{n}^{\left(3\right)}\left(x_{1}\right)-D_{n}^{\left(1\right)}\left(x_{R}\right)\right]}}{\frac{\left[D_{n}^{\left(3\right)}\left(x_{1}\right)-\eta_{r}D_{n}^{\left(1\right)}\left(x_{L}\right)\right]}{\left[\eta_{r}D_{n}^{\left(3\right)}\left(x_{1}\right)-D_{n}^{\left(1\right)}\left(x_{L}\right)\right]}-\frac{\left[\eta_{r}D_{n}^{\left(1\right)}\left(x_{R}\right)-D_{n}^{\left(3\right)}\left(x_{1}\right)\right]}{\left[\eta_{r}D_{n}^{\left(3\right)}\left(x_{1}\right)-D_{n}^{\left(1\right)}\left(x_{R}\right)\right]}}\\ \; & +p_{mn}^{u}\cdot \frac{\psi_{n}\left(x_{1}\right)}{\xi_{n}\left(x_{1}\right)}\frac{\frac{\left[\eta_{r}D_{n}^{\left(1\right)}\left(x_{L}\right)-D_{n}^{\left(1\right)}\left(x_{1}\right)\right]}{\left[\eta_{r}D_{n}^{\left(3\right)}\left(x_{1}\right)-D_{n}^{\left(1\right)}\left(x_{L}\right)\right]}-\frac{\left[D_{n}^{\left(1\right)}\left(x_{1}\right)-\eta_{r}D_{n}^{\left(1\right)}\left(x_{R}\right)\right]}{\left[\eta_{r}D_{n}^{\left(3\right)}\left(x_{1}\right)-D_{n}^{\left(1\right)}\left(x_{R}\right)\right]}}{\frac{\left[D_{n}^{\left(3\right)}\left(x_{1}\right)-\eta_{r}D_{n}^{\left(1\right)}\left(x_{L}\right)\right]}{\left[\eta_{r}D_{n}^{\left(3\right)}\left(x_{1}\right)-D_{n}^{\left(1\right)}\left(x_{L}\right)\right]}-\frac{\left[\eta_{r}D_{n}^{\left(1\right)}\left(x_{R}\right)-D_{n}^{\left(3\right)}\left(x_{1}\right)\right]}{\left[\eta_{r}D_{n}^{\left(3\right)}\left(x_{1}\right)-D_{n}^{\left(1\right)}\left(x_{R}\right)\right]}} \end{array}$$
Correspondingly, substituting Formula (Equation 41) into the internal field expansion coefficients (Equation 38)–(Equation 40), the internal field expansion coefficients are rewritten as(44)$$A_{mn}=\eta_{r}\frac{x_{R}}{x_{1}}\cdot \frac{\psi_{n}\left(x_{1}\right)}{\psi_{n}\left(x_{R}\right)}\frac{\frac{\left[D_{n}^{\left(1\right)}\left(x_{1}\right)-D_{n}^{\left(3\right)}\left(x_{1}\right)\right]}{\left[D_{n}^{\left(1\right)}\left(x_{L}\right)-\eta_{r}D_{n}^{\left(3\right)}\left(x_{1}\right)\right]}\cdot q_{mn}^{u}+\frac{\left[D_{n}^{\left(1\right)}\left(x_{1}\right)-D_{n}^{\left(3\right)}\left(x_{1}\right)\right]}{\left[\eta_{r}D_{n}^{\left(1\right)}\left(x_{L}\right)-D_{n}^{\left(3\right)}\left(x_{1}\right)\right]}\cdot p_{mn}^{u}}{\left\{\frac{\left[D_{n}^{\left(1\right)}\left(x_{R}\right)-\eta_{r}D_{n}^{\left(3\right)}\left(x_{1}\right)\right]}{\left[D_{n}^{\left(1\right)}\left(x_{L}\right)-\eta_{r}D_{n}^{\left(3\right)}\left(x_{1}\right)\right]}+\frac{\left[\eta_{r}D_{n}^{\left(1\right)}\left(x_{R}\right)-D_{n}^{\left(3\right)}\left(x_{1}\right)\right]}{\left[\eta_{r}D_{n}^{\left(1\right)}\left(x_{L}\right)-D_{n}^{\left(3\right)}\left(x_{1}\right)\right]}\right\}}$$(45)$$B_{mn}=\eta_{r}\frac{x_{L}}{x_{1}}\cdot \frac{\psi_{n}\left(x_{1}\right)}{\psi_{n}\left(x_{L}\right)}\frac{\frac{\left[D_{n}^{\left(1\right)}\left(x_{1}\right)-D_{n}^{\left(3\right)}\left(x_{1}\right)\right]}{\left[D_{n}^{\left(1\right)}\left(x_{R}\right)-\eta_{r}D_{n}^{\left(3\right)}\left(x_{1}\right)\right]}\cdot q_{mn}^{u}-\frac{\left[D_{n}^{\left(1\right)}\left(x_{1}\right)-D_{n}^{\left(3\right)}\left(x_{1}\right)\right]}{\left[\eta_{r}D_{n}^{\left(1\right)}\left(x_{R}\right)-D_{n}^{\left(3\right)}\left(x_{1}\right)\right]}\cdot p_{mn}^{u}}{\left\{\frac{\left[\eta_{r}D_{n}^{\left(1\right)}\left(x_{L}\right)-D_{n}^{\left(3\right)}\left(x_{1}\right)\right]}{\left[\eta_{r}D_{n}^{\left(1\right)}\left(x_{R}\right)-D_{n}^{\left(3\right)}\left(x_{1}\right)\right]}+\frac{\left[D_{n}^{\left(1\right)}\left(x_{L}\right)-\eta_{r}D_{n}^{\left(3\right)}\left(x_{1}\right)\right]}{\left[D_{n}^{\left(1\right)}\left(x_{R}\right)-\eta_{r}D_{n}^{\left(3\right)}\left(x_{1}\right)\right]}\right\}}$$

### 2.3. Far-Field Scattering of a Bessel Pincer Light-Sheet Beam on a Chiral Sphere

For a chiral dielectric sphere, when r≫λ, the asymptotic behavior of the vector spherical harmonic functions remains unchanged; that is,(46)$$\begin{array}{cc} M_{mn}^{\left(j\right)}\left(k,r\right) & =\left[i\pi_{mn}\left(\cos\theta \right)e_{\theta }-\tau_{mn}\left(\cos\theta \right)e_{\phi }\right]z_{n}^{\left(j\right)}\left(kr\right)\exp\left(im\phi \right)\\ \; & \approx \left[\pi_{mn}\left(\cos\theta \right)e_{\theta }+i\tau_{mn}\left(\cos\theta \right)e_{\phi }\right]\frac{1}{kr}{\left(-i\right)}^{n}\exp\left(ikr\right)\exp\left(im\phi \right) \end{array}$$(47)$$\begin{array}{cc} N_{mn}^{\left(j\right)}\left(k,r\right) & =\left[\tau_{mn}\left(\cos\theta \right)e_{\theta }+i\pi_{mn}\left(\cos\theta \right)e_{\phi }\right]\frac{1}{kr}\frac{d}{dr}\left[rz_{n}^{\left(j\right)}\left(kr\right)\right]\exp\left(im\phi \right)\\ \; & \;+e_{r}n\left(n+1\right)P_{n}^{m}\left(\cos\theta \right)\frac{z_{n}^{\left(j\right)}\left(kr\right)}{kr}\exp\left(im\phi \right)\\ \; & \approx \left[\tau_{mn}\left(\cos\theta \right)e_{\theta }+i\pi_{mn}\left(\cos\theta \right)e_{\phi }\right]\frac{1}{kr}{\left(-i\right)}^{n}\exp\left(ikr\right)\exp\left(im\phi \right) \end{array}$$

Substituting the above two equations into Equation (Equation 30), the representation of the far-field scattering intensity is similar to that of an achiral dielectric sphere, expressed as(48)$$\begin{array}{cc} E_{sca} & =\frac{\exp(ikr)}{kr}\sum_{n=1}^{\infty }\sum_{m=-n}^{n}iE_{mn}{\left(-i\right)}^{n}\cdot \exp\left(im\phi \right)\\ \; & \times \left\{\begin{array}{c} \left[A_{mn}^{s}\pi_{mn}\left(\cos\theta \right)+B_{mn}^{s}\tau_{mn}\left(\cos\theta \right)\right]e_{\theta }\\ +\left[A_{mn}^{s}i\tau_{mn}\left(\cos\theta \right)+B_{mn}^{s}i\pi_{mn}\left(\cos\theta \right)\right]e_{\phi } \end{array}\right\} \end{array}$$

The spherical coordinate components have only θ and ϕ components, and their scalar forms are expressed as(49)$$\begin{array}{cc} E_{\theta } & =\frac{i}{kr}\exp\left(ikr\right)\sum_{n=1}^{\infty }\sum_{m=-n}^{n}E_{mn}{\left(-i\right)}^{n}\exp\left(im\phi \right)\\ \; & \times \left[A_{mn}^{s}\pi_{mn}\left(\cos\theta \right)+B_{mn}^{s}\tau_{mn}\left(\cos\theta \right)\right]\\ \; & =\frac{i}{kr}\exp\left(ikr\right)S_{\theta } \end{array}$$(50)$$\begin{array}{cc} E_{\phi } & =-\frac{1}{kr}\exp\left(ikr\right)\sum_{n=1}^{\infty }\sum_{m=-n}^{n}E_{mn}{\left(-i\right)}^{n}\exp\left(im\phi \right)\\ \; & \times \left[A_{mn}^{s}\tau_{mn}\left(\cos\theta \right)+B_{mn}^{s}\pi_{mn}\left(\cos\theta \right)\right]\\ \; & =-\frac{1}{kr}\exp\left(ikr\right)S_{\phi } \end{array}$$

Here, $S_{\theta }$ and $S_{\phi }$ are the general scattering intensity functions, expressed as(51)$$S_{\theta }=\sum_{n=1}^{\infty }\sum_{m=-n}^{n}E_{mn}{\left(-i\right)}^{n}\left[A_{mn}^{s}\pi_{mn}\left(\cos\theta \right)+B_{mn}^{s}\tau_{mn}\left(\cos\theta \right)\right]\exp\left(im\phi \right)$$(52)$$S_{\phi }=\sum_{n=1}^{\infty }\sum_{m=-n}^{n}E_{mn}{\left(-i\right)}^{n}\left[A_{mn}^{s}\tau_{mn}\left(\cos\theta \right)+B_{mn}^{s}\pi_{mn}\left(\cos\theta \right)\right]\exp\left(im\phi \right)$$

## 3. Results

Initially, the numerical results are developed by a Python program to compute the distribution of field intensities (including scattered-field ${\left|E_{sca}^{u}\right|}^{2}$; near-surface-field ${\left|E_{tol}^{u}\right|}^{2}$, where $E_{tol}^{u}=E_{sca}^{u}+E_{inc}^{u}+E_{\mathrm{int}}^{u}$; and far-field intensities ($I=S_{\theta }+S_{\varphi }$)). We chiefly pay attention to the beam parameters (scaling parameter $\alpha_{0}$ and order *l* of Bessel pincer light sheet) and particle parameters (dimensionless particle parameter ka, and dimensionless chirality parameter κ). For the convenience of numerical calculation, we assume that the position of the beam center relative to the particle is at (0,0,−10)μm, while the center of the sphere is located at (0,0,0)μm. The refractive index of the chiral particle is $m_{1}=1.33+0.01i$, and the refractive index of the surrounding medium is $m_{2}$ = 1.0, and the dimensionless chirality parameter κ is hereinafter referred to as the chirality parameter. The wavelength of the incident beam in the surrounding medium is λ=0.6328μm, and the equivalent radius of the chiral sphere is denoted by the symbol *a*. In the calculation of the field distribution, the amplitude of the electric field is defined as $E_{0}=1$ V/m. The intensity mentioned in this paper is the relative intensity ${\left|E\right|}^{2}/{\left|E_{0}\right|}^{2}$.

### 3.1. Scattering Field Distribution of a Chiral Sphere Under the Action of Incident Beam

The illustrations presented in Figure 3 demonstrate the distributions of scattered fields within the yoz plane, resulting from a Bessel pincer light-sheet beam interacting with a chiral particle. This analysis investigates the modulation of scattering as influenced by the chiral parameter κ (with values of [0,0.1,−0.1]) under the conditions of LCP (illustrated in Figure 3a,c,e) and RCP (illustrated in Figure 3b,d,f) light. For the case where κ=0, as shown in figures (a) and (b), the scattered-field distributions of the chiral sphere under the action of LCP and RCP show symmetrical characteristics. When κ=0.1, as shown in figures (c) and (d), the scattered-field distributions corresponding to LCP and RCP change, and there is a mirror symmetry relationship between them. This symmetrical relationship is caused by the combined effect of the change in the chirality parameter κ and the polarization characteristics of the beam. In the scenario where κ=−0.1, as shown in figures (e) and (f), the distribution of the scattered field changes again, and the scattered fields under the action of LCP and RCP still maintain mirror symmetry. This indicates that the positive or negative value of the chirality parameter κ affects the distribution of the scattered field, and the mirror symmetry is the result of the interaction between the chirality and polarization characteristics. The positive or negative values of the chirality parameter κ significantly affect the scattered-field distribution of the chiral sphere under the action of LCP and RCP beams. This effect is closely related to the chirality and polarization characteristics. At the same time, the mirror symmetry shown by the scattered-field distribution further confirms the internal relationship between the chirality and polarization characteristics.

Figure 4 depicts the scattered-field intensity distribution of a chiral sphere illuminated by an RCP Bessel pincer light-sheet beam as the particle radius *a* increases from 0.1λ to 3λ. For the Rayleigh regime (a=0.1λ≪λ), the scattered-field intensity is approximately isotropic (Figure 4a), showing a uniform annular weak-intensity distribution in the yoz plane. Dipole-dominated scattering conforms to Rayleigh characteristics, with intensity proportional to the square of particle volume and inversely proportional to $\lambda^{4}$. In the Mie transition and development stage (a=0.5λ→3λ): At a=0.5λ (Figure 4b), scattering deviates from isotropy, exhibiting asymmetry with local intensity enhancement (e.g., positive z-direction), marking the onset of Mie scattering due to particle interference and diffraction. For a=1λ→3λ (Figure 4c–f), spatial asymmetry intensifies. The strong-field region concentrates directionally (e.g., along the z-axis), with a significant increase in amplitude. Mie scattering dominates, driven by intensified multipole interactions (such as dipole, quadrupole, etc.), resulting in a complex angular dependence. The strong-field “wake” extends with increasing *a*, reflecting enhanced directionality and local field enhancement as *a* becomes comparable to λ. In summary, the scattered-field intensity evolves from an “isotropic weak distribution” (Rayleigh) through an “asymmetric enhancement” (transition) to a “directional strong-field concentration” (Mie). Fundamentally, the a/λ ratio drives the transition from Rayleigh dipole to Mie multipole scattering, transforming the field from uniform to highly directional with increased amplitude.

Figure 5 shows scattered-field distributions in the yoz plane for a spherical particle (white circle) illuminated by an LCP Bessel pincer light-sheet beam, evolving with beam scaling parameter $\alpha_{0}$. At $\alpha_{0}=0.3$ (Figure 5a), a striking local enhancement concentrates on one particle side (positive z-direction), with a peak intensity of 0.6, reflecting focused energy channeling. Increasing the beam scaling parameter to $\alpha_{0}=0.5$ (Figure 5b) broadens the strong-field region while reducing the amplitude (peak 0.20), indicating transition toward uniform distribution. At $\alpha_{0}=0.7$ (Figure 5c), the amplitude further decreases (peak 0.10), with narrowed concentration and gentler gradients, demonstrating dispersed energy redistribution. Thus, $\alpha_{0}$ critically controls the scattered-field intensity distribution: smaller values ($\alpha_{0}=0.3$) enable localized field focusing for strong-field applications, while larger values ($\alpha_{0}=0.7$) promote dispersion and intensity reduction for broad modulation.

The figures presented in Figure 6 are the same as Figure 5 but with different beam orders *l*. For l=1 (Figure 6a), the scattered field exhibits moderate intensity, with strong-field regions concentrated on one side of the sphere (e.g., positive z-direction) and a gentle intensity gradient, indicating limited energy concentration due to low-order beam interactions. At l=3 (Figure 6b), the scattered-field intensity is notably enhanced. Strong-field regions intensify and become more concentrated along the z-axis, reflecting improved light–matter coupling efficiency driven by the beam’s higher-order topological structure. With l=5 (Figure 6c), intensity reaches its highest level. Strong-field regions remain concentrated on one side, exhibiting a steeper intensity gradient, signifying energy confinement within a narrower spatial range. This demonstrates amplified topological effects, enabling precise control over the light–chiral sphere interaction. In summary, as *l* increases from 1 to 5, the scattered-field intensity continuously increases and the spatial concentration of strong-field regions intensifies. This evolution stems from the increasingly complex topological structure of the incident field at higher *l*, strengthening light–matter interactions with the chiral sphere.

### 3.2. Near-Surface Field

Figure 7 shows near-surface scattered-field distributions in the yoz plane for a chiral particle illuminated by a Bessel pincer light-sheet beam, demonstrating κ modulation ([0, 0.1, −0.1]) under LCP (a,c,e) and RCP (b,d,f) illumination. At κ=0 (achiral reference), both polarizations yield symmetric periodic patterns with minimal polarization-induced asymmetry. For κ=0.1 (positive chirality), LCP produces asymmetric scattering with enhanced complexity and forward-scattering gradients, while RCP shows polarization-specific redistribution with shifted/broadened patterns. At κ=−0.1 (negative chirality), LCP/RCP exhibit inverted asymmetries relative to κ=0.1. Non-zero κ thus breaks near-field symmetry for both polarizations, with modulation dependent on the chirality sign (±κ) and incident polarization (LCP/RCP)—reflecting distinct spin–angular momentum interactions. The symmetric κ=0 case confirms that chirality drives polarization-specific alterations. These results highlight κ’s ability to asymmetrically tune near-surface scattering between LCP/RCP, essential for designing chiral optical devices through polarization-dependent control.

Furthermore, given the mirror symmetry of chiral particles in both right-circularly polarized (RCP) and left-circularly polarized (LCP) modes, subsequent discussions concerning the near-surface field characteristics of chiral particles—particularly concerning variations in particle size and beam parameters—can be limited to a single polarization state of the incident beam.

The study in Figure 8 examines the near-surface scattered-field intensity changes when a chiral (accounting for chirality, κ=0.1) and non-chiral (ignoring chirality, κ=0) sphere are illuminated by an LCP Bessel pincer light-sheet beam, with the particle radius *a* increasing from 2λ to 2.2λ. Overall, for both sphere types under LCP illumination, increasing the angle generally boosts the near-surface scattered-field intensity. For non-chiral spheres (κ=0), as a grows from 2λ (Figure 8a) to 2.1λ (Figure 8b) and then to 2.2λ (Figure 8c), the near-surface field intensity rises. The spatial distribution becomes more concentrated, showing a relatively “regular” evolution. For chiral spheres (κ=0.1), at a=2λ (Figure 8d) the near-surface field has distinct intensity and distribution features due to chirality. As *a* increases to 2.1λ (Figure 8e) and 2.2λ (Figure 8f), the field intensity also increases. Still, chirality introduces additional interaction mechanisms, leading to more complex evolution of the near-surface field compared to non-chiral spheres.

The graphs presented in Figure 9 are the same as in Figure 8 but with varying the beam scaling parameter $\alpha_{0}$. For non-chiral spheres (κ = 0), larger $\alpha_{0}$ values enhance LCP beam interaction and alter the near-surface field: at $\alpha_{0}$ = 0.3 (Figure 9a), specific intensity regions emerge; at $\alpha_{0}$ = 0.5 (Figure 9b), intensity increases with growing spatial complexity; and at $\alpha_{0}$ = 0.7 (Figure 9c), overall intensity and internal structure evolve further. In contrast, chiral spheres (κ = 0.1) exhibit distinct chirality-driven evolution: at $\alpha_{0}$ = 0.3 (Figure 9d), intensity/distribution diverges markedly from non-chiral cases; at $\alpha_{0}$ = 0.5 (Figure 9e), intensity increases with unique spatial modulation; and at $\alpha_{0}$ = 0.7 (Figure 9f), significant intensity changes and intricate internal structures develop due to chirality-specific interaction mechanisms. Thus, while increasing $\alpha_{0}$ modifies near-surface fields for both sphere types under LCP illumination, non-chiral spheres show regular evolution, whereas chiral spheres exhibit complex, chirality-modulated changes.

The graphs presented in Figure 10 are the same as in Figure 8 but with different beam orders *l*. For non-chiral spheres (κ = 0), increasing *l* enhances LCP beam interaction and alters the near-surface field: at *l* = 1 (Figure 10a), specific intensity regions emerge; at *l* = 5 (Figure 10b), intensity rises with growing spatial complexity; and at *l* = 11 (Figure 10c), overall intensity and internal structure evolve further. Conversely, chiral spheres (κ = 0.1) exhibit distinct chirality-driven evolution: at *l* = 1 (Figure 10d), intensity/distribution diverges from non-chiral cases; at *l* = 5 (Figure 10e), intensity increases with unique spatial modulation; and at *l* = 11 (Figure 10f), significant intensity changes and elaborate internal structures develop due to chirality-specific interaction mechanisms. Thus, while increasing *l* modifies near-surface fields for both sphere types, non-chiral spheres show “regular” evolution, whereas chiral spheres exhibit complex, chirality-modulated changes.

### 3.3. Far-Region Scattered Field

Figure 11 shows far-field scattered intensity distributions in Cartesian coordinates under LCP illumination (Figure 11a,b), exploring modulation of the chiral parameter κ ([−0.1, 0, 0.1]). For κ = 0 (non-chiral), both polarizations exhibit symmetric distributions with peaks at $\left|\theta \right|\leq 30^{\circ }$, showing minimal polarization dependence. With κ = 0.1 (positive chirality), LCP displays enhanced forward scattering (higher $\theta =0^{\circ }$ peak) with suppressed side-lobes (e.g., $\theta =\pm 40^{\circ }$), while RCP redistributes asymmetrically, with distinct peak shifts and side-lobe modifications. For κ = −0.1 (negative chirality), LCP inverts the κ = 0.1 response (weakened forward peak, enhanced side-lobes), whereas RCP shows polarization-specific inverted modulation relative to κ = 0.1. Thus, a non-zero κ breaks far-field symmetry for both polarizations, with the modulation direction being polarization-dependent. LCP and RCP respond asymmetrically to ±κ, reflecting differential spin–angular momentum interactions. The symmetric κ = 0 case confirms that chirality drives polarization-specific changes. These results demonstrate the ability of κ to asymmetrically tune LCP/RCP far-field scattering, essential for designing chiral optical devices through polarization-dependent pattern control.

Figure 12 shows far-field scattering patterns in polar coordinates for a particle illuminated by a Bessel pincer light-sheet beam, exploring modulation of particle size (ka = [2,5,8]) and chirality (κ = [−0.1,0.1]) under LCP (Figure 12a) and RCP (Figure 12b) illumination. At ka = 2 (small particle), both polarizations exhibit broad, low-intensity lobes, with ±κ causing subtle positional shifts. For ka=5, LCP develops narrower lobes, where κ>0 enhances forward scattering while κ<0 suppresses it; RCP shows distinct chiral modulation with stronger asymmetry. At ka = 8, LCP forms multi-lobe patterns that are drastically reshaped by ±κ, while RCP yields greater complexity, dominated by chirality–polarization coupling. Thus, increasing the particle size (ka) evolves scattering from broad lobes to structured multi-lobe patterns while amplifying κ-induced asymmetry. Crucially, ±κ induces polarization-specific modulation—enhanced at larger ka—with LCP/RCP exhibiting distinct patterns due to differential spin–angular momentum coupling. These results demonstrate synergistic control of far-field scattering via ka and κ for designing chirality-sensitive optical systems.

Figure 13 shows far-field scattering patterns in polar coordinates for a chiral particle illuminated by a Bessel pincer light-sheet beam, as the scaling parameter $\alpha_{0}$ ([0.5,0.6,0.7]) modulates scattering under LCP (Figure 13a) and RCP (Figure 13b) illumination. At $\alpha_{0}=0.5$ (narrower beam), both polarizations show compact lobes, with ±κ inducing subtle asymmetry and polarization-specific phase shifts. Increasing to $\alpha_{0}=0.6$ broadens the lobes and enhances the chiral effects: LCP exhibits κ-dependent forward/side-lobe modulation, while RCP shows amplified lobe splitting. At $\alpha_{0}=0.7$ (wider beam), scattering develops extended multi-lobed patterns: LCP demonstrates drastic κ-reshaped distributions, whereas RCP produces extreme lobe complexity with unique asymmetry. Thus, a larger $\alpha_{0}$ transforms scattering from compact lobes to extended, asymmetric patterns by enhancing the light–particle interaction range, thereby amplifying chiral asymmetry. Crucially, ±κ induces polarization-specific modulation—more pronounced at higher $\alpha_{0}$—with distinct LCP/RCP dynamics, reflecting their spin–angular momentum coupling differences.

Figure 14 is the same as Figure 13 but with different beam orders *l*. For l=1 (low-order beam, simple mode), for LCP, the far-field pattern shows broad, low-contrast lobes with minimal angular complexity, and the chiral parameter ±κ induces subtle asymmetry; for RCP, similar broad, symmetric lobes appear, yet with polarization-specific phase shifts under ±κ. When l=3 (medium-order beam, increased mode complexity), for LCP, scattering develops narrower, higher-contrast lobes, with positive κ sharpening forward-scattering peaks and creating new side-lobes while negative κ broadens existing lobes, enhancing chirality-induced angular asymmetry; for RCP, it amplifies polarization-specific modulation, with chiral effects showing stronger lobe splitting under ±κ with RCP. At l=5 (high-order beam, complex mode), for LCP, scattering produces highly structured, multi-lobed patterns, with positive/negative κ drastically reshaping lobe distributions; for RCP, it leads to extreme lobe complexity, with the chirality–polarization interaction dominating and showing unique lobe asymmetry compared to LCP under ±κ.

## 4. Conclusions

This study investigates the scattering characteristics of circularly polarized Bessel pincer light sheets interacting with a chiral sphere of arbitrary size within the framework of the generalized Lorenz–Mie theory (GLMT). Using the vector angular spectrum method, series expansions for the incident, internal, and scattered fields are derived using the Condon–Rosenfeld relations and spherical vector wave functions (SVWFs). Detailed analyses of the scattered-field, near-field, and far-field scattering characteristics are conducted to elucidate the influence of key beam and particle parameters.

Firstly, numerical simulations reveal that the beam scaling factor ($\alpha_{0}$), beam order (*l*), and chirality parameter (κ) have a significant impact on both near-field and far-field scattering. The chiral parameter κ induces polarization-dependent asymmetry in the scattered field, with its sign (±) dictating the modulation direction for left-circularly polarized (LCP) versus right-circularly polarized (RCP) light. The beam scaling parameter $\alpha_{0}$ governs the near-to-far-field transition: a smaller $\alpha_{0}$ enhances near-field polarization effects and local intensity, while a larger $\alpha_{0}$ promotes a uniform far-field distribution with reduced polarization sensitivity. Higher beam orders (*l*) intensify the scattering strength and clarify the distribution. Collectively, κ enables asymmetric tuning, $\alpha_{0}$ controls spatial confinement, and *l* amplifies light–particle interactions, providing versatile design strategies for chiral photonic devices. Then, a non-zero κ fundamentally breaks scattering symmetry and dictates polarization-selective asymmetry, with its sign governing directional asymmetry under LCP/RCP illumination. Increasing the particle size (characterized by $\alpha_{0}$/ka), beam order (*l*), or scaling parameter ($\alpha_{0}$) systematically amplifies near-field complexity and asymmetry, irrespective of chirality. Crucially, larger particle radius, *l*, or $\alpha_{0}$ values synergistically enhance the κ-mediated polarization dichotomy—intensifying chiral-light interactions and significantly differentiating RCP/LCP responses, even at identical parameter ratios. These parameters collectively govern spin–angular momentum coupling, enabling precise nanoscale control over polarization-dependent near-field scattering. The chiral parameter κ also fundamentally breaks far-field scattering symmetry for both LCP and RCP illumination, inducing polarization-dependent intensity modulation: ±κ generates asymmetric, often mirrored, redistributions (e.g., forward peak enhancement/suppression, side-lobe alterations), directly reflecting differential spin–angular momentum coupling. Particle size (ka) critically amplifies these chiral effects: increasing ka transitions scattering from broad, weakly modulated lobes to intense, multi-lobed patterns with significantly amplified κ-induced asymmetry and pronounced polarization contrast. Similarly, a larger $\alpha_{0}$ or higher *l* intensifies chirality–polarization interactions, evolving scattering patterns from compact/simple distributions to extended, highly structured lobes with drastically reshaped, polarization-specific asymmetry under ±κ. Critically, LCP and RCP exhibit distinct and frequently opposing responses to identical κ across all parameters (ka, $\alpha_{0}$, *l*), while increased ka, $\alpha_{0}$, or *l* universally enhances the magnitude and complexity of κ’s polarization-selective modulation. Thus, κ, ka, $\alpha_{0}$, and *l* synergistically control polarization-tailored far-field patterns.

This work demonstrates synergistic control of the chiral parameter κ, particle size ka, beam scaling factor $\alpha_{0}$, and beam order *l* over polarization-selective near- and far-field scattering from chiral spheres under Bessel beam illumination. Future work should extend these findings to optical trapping and tweezers, enabling advanced manipulation of chiral particles and the development of spin-dependent trapping systems.

## Figures and Tables

**Figure 1 micromachines-16-00845-f001:**
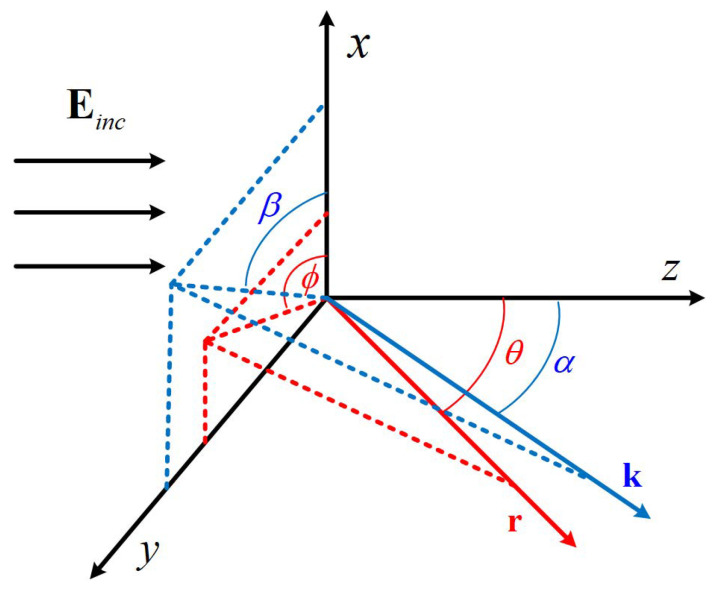
Schematic diagram of propagation-angle dependence of circularly polarized electromagnetic waves.

**Figure 2 micromachines-16-00845-f002:**
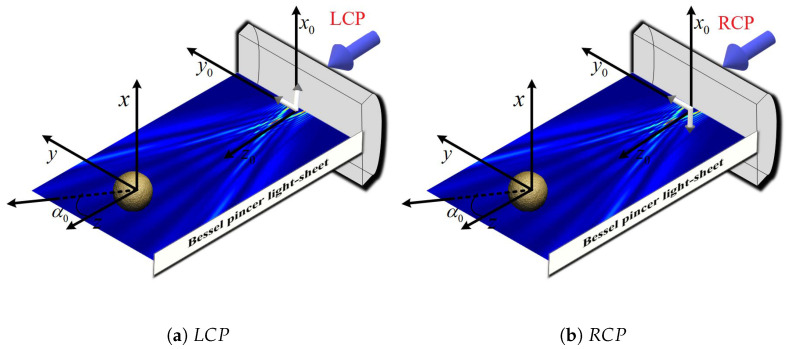
Graphical representation for the interaction of a circularly polarized Bessel pincer light-sheet beam with a chiral sphere. (**a**) Left-handed circular polarization labeled as $\left({\hat{e}}_{y}+i{\hat{e}}_{x}\right)$; (**b**) right-handed circular polarization labeled as $\left({\hat{e}}_{y}-i{\hat{e}}_{x}\right)$.

**Figure 3 micromachines-16-00845-f003:**
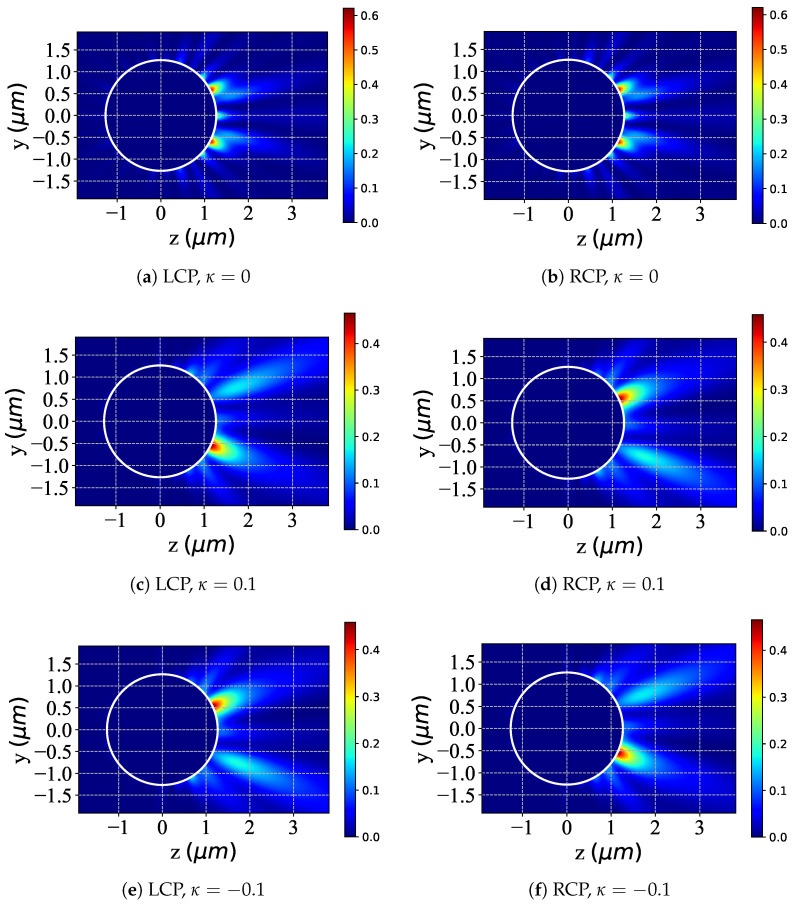
Distribution of the scattered field in the yoz plane under different values of chirality parameter κ. λ=0.6328μm, a=2λ, $\epsilon_{r}=1.7688-0.0268i$, $\mu_{r}=1$, and κ=[0,0.1,−0.1]; beam order l=10, beam scaling parameter $\alpha_{0}=0.5$.

**Figure 4 micromachines-16-00845-f004:**
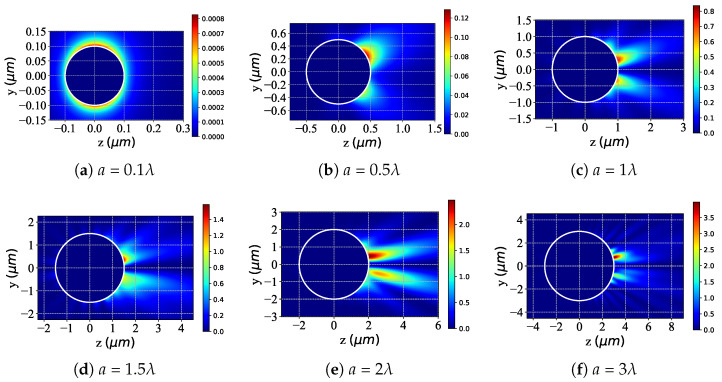
Distribution of the scattered field in the yoz plane with different dimensionless particle radius *a* under illumination of an RCP Bessel pincer light-sheet beam. λ=0.6328μm, $\epsilon_{r}=1.7688-0.0268i$, $\mu_{r}=1$, κ=0.1, and a=[0.1λ,0.5λ,1λ,1.5λ,2λ,3λ]; beam order l=1, beam scaling parameter $\alpha_{0}=0.25$.

**Figure 5 micromachines-16-00845-f005:**
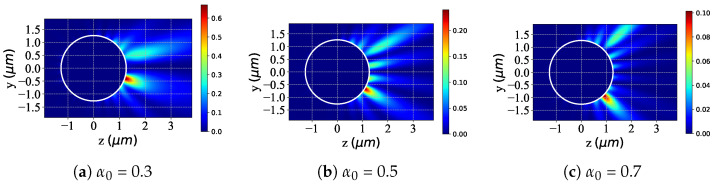
Distribution of the scattered field in the yoz plane with different values of beam scaling parameter $\alpha_{0}$ under LCP beam illumination. λ=0.6328μm, a=2λ, $\epsilon_{r}=1.7688-0.0268i$, $\mu_{r}=1$, and κ=0.1; beam order l=1 and beam scaling parameter $\alpha_{0}$ = [0.3, 0.5, 0.7].

**Figure 6 micromachines-16-00845-f006:**
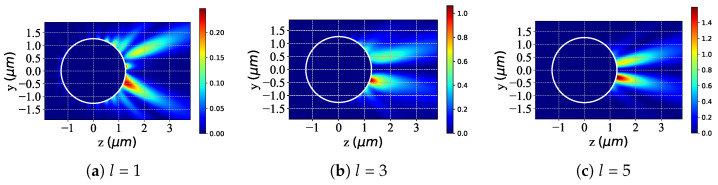
The same as Figure 5 but with different beam orders *l*. λ=0.6328μm, a=2λ, $\epsilon_{r}=1.7688-0.0268i$, $\mu_{r}=1$, κ=0.1, $\alpha_{0}=0.34$, and l=[1,3,5].

**Figure 7 micromachines-16-00845-f007:**
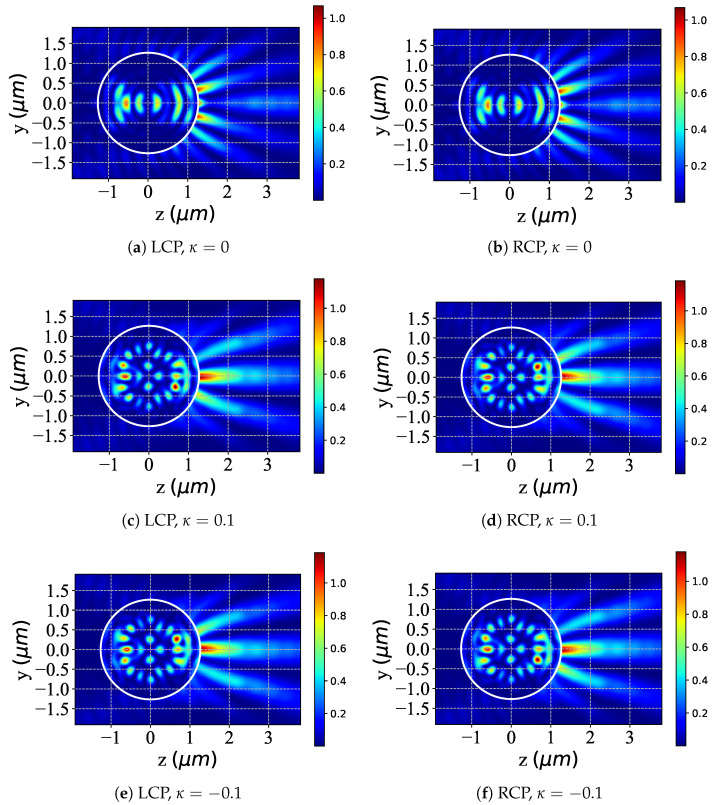
Distribution of the near-surface scattered field with different values of chirality parameter κ in the yoz plane. λ=0.6328μm, a=2λ, $\epsilon_{r}=1.7688-0.0268i$, $\mu_{r}=1$, and κ=[0,0.1,−0.1]; beam order l=10, beam scaling parameter $\alpha_{0}=0.42$.

**Figure 8 micromachines-16-00845-f008:**
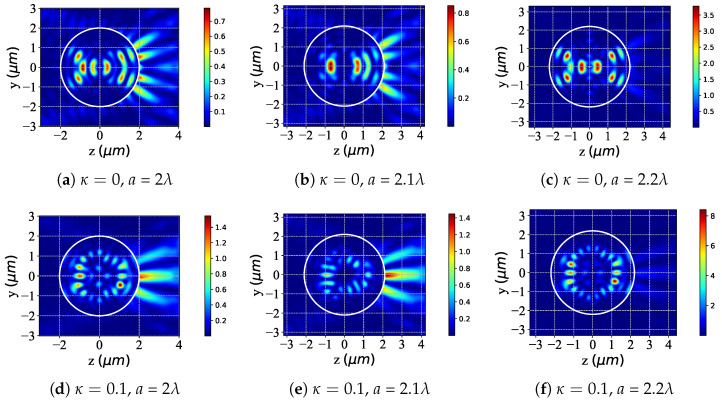
Distribution of the near-surface scattered field with different values of particle radius *a* under LCP beam illumination. λ=0.6328μm, $\epsilon_{r}=1.7688-0.0268i$, $\mu_{r}=1$, κ=[0,0.1], and a=[2λ,2.1λ,2.2λ]; beam order l=1, beam scaling parameter $\alpha_{0}=0.25$.

**Figure 9 micromachines-16-00845-f009:**
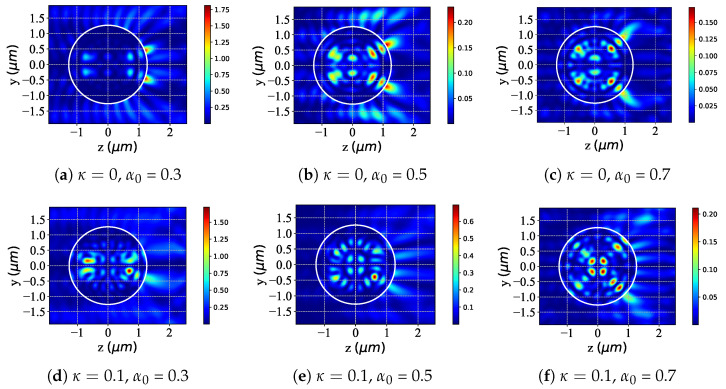
The same as Figure 8 but with different values of beam scaling parameter $\alpha_{0}$ in the yoz plane. λ=0.6328μm, a=2λ, $\epsilon_{r}=1.7688-0.0268i$, $\mu_{r}=1$, and κ=[0,0.1]. Beam order l=1 and $\alpha_{0}=\left[0.3,0.5,0.7\right]$.

**Figure 10 micromachines-16-00845-f010:**
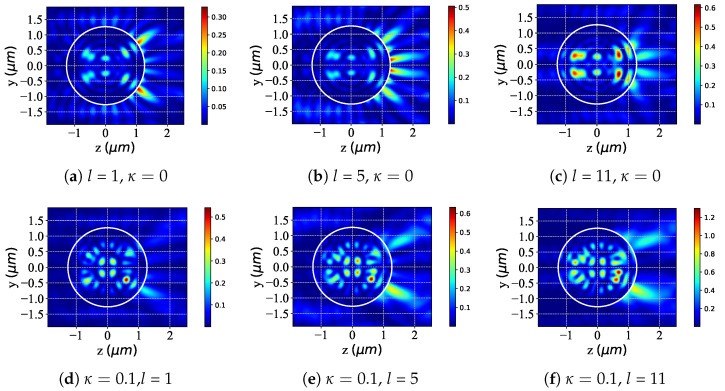
The same as Figure 8 but with different beam orders *l* in the yoz plane. λ=0.6328μm, $\epsilon_{r}=1.7688-0.0268i$, $\mu_{r}=1$, κ=[0,0.1], $\alpha_{0}=0.54$, and l=[1,5,11].

**Figure 11 micromachines-16-00845-f011:**
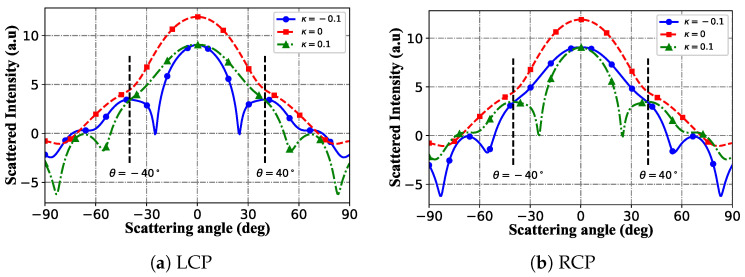
Distribution of the far field in Cartesian coordinates. λ=0.6328μm, a=λ, $\epsilon_{r}=1.7688-0.0268j$, $\mu_{r}\;=\;1$, κ=[−0.1,0,0.1], $\alpha_{0}=0.2$, and l=2.

**Figure 12 micromachines-16-00845-f012:**
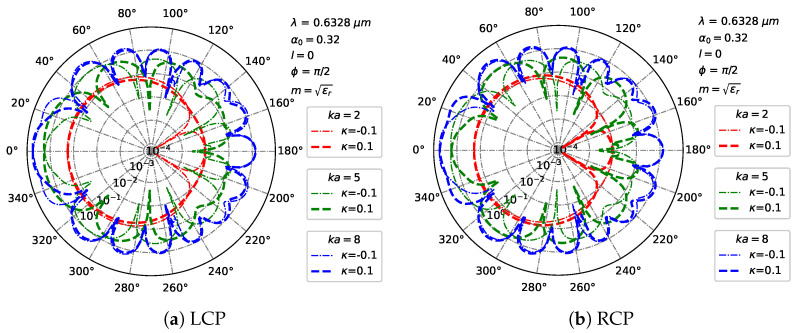
Distribution of the far field in Cartesian coordinates under different beam orders *l* of Bessel pincer light-sheet beam. λ=0.6328μm, a=λ, $\epsilon_{r}=1.7688-0.0268i$, $\mu_{r}=1$, κ=[−0.1,0.1], $\alpha_{0}=0.32$, and l=0.

**Figure 13 micromachines-16-00845-f013:**
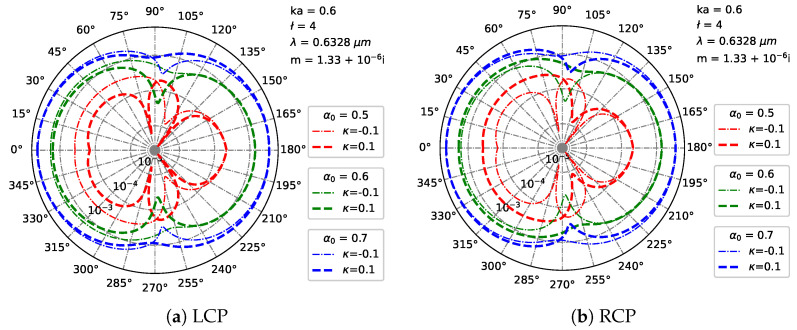
Distribution of the far field in Cartesian coordinates under different beam orders *l* of Bessel pincer light-sheet beam. λ=0.6328μm, a=λ, $\epsilon_{r}=1.7688-0.0268i$, $\mu_{r}=1$, κ=[−0.1,0.1], $\alpha_{0}=\left[0.5,0.6,0.7\right]$, and l=4.

**Figure 14 micromachines-16-00845-f014:**
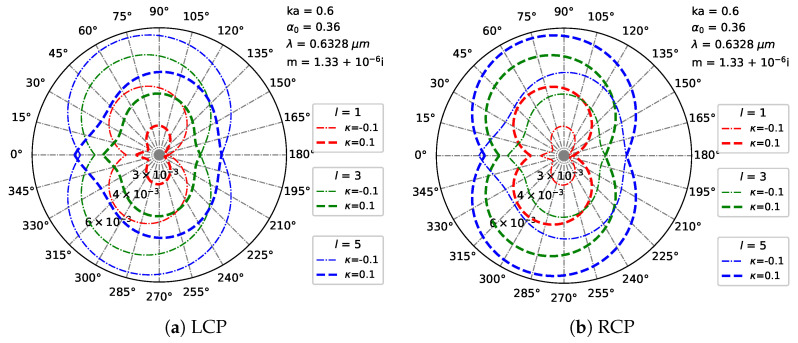
Distribution of the far field in Cartesian coordinates under different beam orders *l* of Bessel pincer light-sheet beam. λ=0.6328μm, a=λ, $\epsilon_{r}=1.7688-0.0268i$, $\mu_{r}=1$, κ=[−0.1,0.1], $\alpha_{0}=0.36$, and l=[1,3,5].

## Data Availability

The original contributions presented in this study are included in the article. Further inquiries can be directed to the corresponding author.

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
