# Peer review of "Scattering Characteristics of a Circularly Polarized Bessel Pincer Light-Sheet Beam Interacting with a Chiral Sphere of Arbitrary Size"

_micromachines, 2025, doi:10.3390/mi16080845_

Round 1

Reviewer 1 Report

Comments and Suggestions for Authors

The scattering interaction between a TE-polarized Bessel pincers light-sheet and a chiral particle  was investigated. Applying the boundary conditions to the chiral sphere, together with the previously obtained expressions for the beam shape coefficients, the values of the incident, internal and scattered fields were obtained using vector spherical wave functions (VSWFs) and multipole expansion. This study reveals significant differences in the scattering characteristics of a linearly-polarized Bessel pincers light-sheet on chiral particles compared to isotropic spheres. First of all, based on previous works, the authors gave a brief introduction to the scattering of a beam by a chiral sphere. It has been shown that the internal field of a chiral sphere can be expanded in terms of VSWFs. Substituting the expression of VSWFs and the asymptotic approximation of Henkel functions, the expression of the scattered field in the far field also can be expressed. Substituting the expression for VSWFs and the asymptotic approximation of Henkel functions, the authors also obtained an expression for the scattered field in the far field. A detailed analysis of the near- and far-field scattering characteristics was carried out in order to understand the effect of beam and particle parameters on scattering. Numerical simulations performed in this study show that parameters such as the beam scaling factor, beam order, and chirality significantly affect the scattering characteristics in both the near and far fields. It has been shown that the distribution of the scattering field undergoes significant changes when the chirality parameter deviates from zero. This deviation introduces asymmetry and leads to complex localized changes in intensity. As the chirality parameter increases, the scattering field becomes more complex and is characterized by pronounced fluctuations and localized intensity peaks. It has been shown that the chirality parameter (K) also plays a crucial role: positive values of K enhance scattering in the forward direction, while negative values enhance scattering in the reverse direction. These effects are crucial for the design and control of scattering behavior in engineering applications. The results demonstrate that this parameter can be effectively used to modulate scattering behavior, providing theoretical information and practical recommendations for developing specific scattering control strategies in engineering applications. These results provide insight into the role of chirality in modulating scattering asymmetry and localization effects.

The article undoubtedly deserves to be published in the journal Micromachines.

Comments on the Quality of English Language

The English could be improved to more clearly express the research.

Reviewer 2 Report

Comments and Suggestions for Authors

The authors investigate scattering of electromagnetic radiation from a chiral sphere.  They give a thorough discussion of their numerical methods and they validate their numerical methods against earlier results, then go on to treat a large number of cases.  (However, see the comment on line 107 below, as there may be an error in the paper.)

Scattering isn't the same as particle manipulation, so there is a disconnect between the title and the paper itself.  This is particularly important given that the journal title is "Micromachines".  If there is just electromagnetic scattering, it's not clear how this is a machine.

There isn't much motivation for why the particular incident field of TE-polarized Bessel pincers was chosen.  Although the authors say in their abstract that there are applications to "single-molecule microbiology", they do not return to this application in the paper.

The results are discussed regarding the sign of kappa.  I would have through that the sign would need to be defined relative to incident left or right circularly polarized light.  Usually, when chirality is discussed, there is a symmetry in that if the chirality of the particle is reversed and RCP and LCP are reversed, the interactions are the same.  While left and right circular polarization appear in the manuscript a little, but it is not clear how they are related to the incident beam which is only described as "TE".  There isn't a diagram of the coordinate system showing the directions of propagation and polarization, although Fig. 1 is an effort in this direction.

The units used in the figure are unclear.  Presumably some fraction of the incident light will scatter, and part of that is present at a given position.

Certain acronyms are not defined such as TE, VSWF, GLMT.  Also yoz. Is o the origin or something else?

Eq. (1).  Why is S(p,q) the notation, rather than S(alpha)?   p and q are not independent, and q does not appear in Eq. (2).

The authors discuss asymmetry, but it is not evident from the figures.  Also, the chirality is the theme of the paper, but there is very little about left and right circularly polarized light.

Another weakness of the paper is that there is no attempt to relate the results to experiment or applications.

Line-by-line comments.

Title.  Pincers

83.  space

87. "angle of propagation"  Compared to what reference?

92. Need a space.

Fig. 1 caption.  The left and center/right parts of the diagram may not have the same scale.   The authors should comment.

99. space

107.  Here, there should be a discussion of left and right circular polarization.  In any case, the formula appears to be incorrect.  I find the square root of the product eigenvalues of the matrix in Eq. (5) to be sqrt( epsilon_r mu_r - kappa^2 ).  At a minimum, the authors should provide a derivation.

133.  "... are easy to diverge ..."  please rephrase.  Consider "often diverge in practice".

140.  closest

142.  need spaces

169.  Formula does not fit on the page.

172.  Reference missing after "Ref."

Fig. 11.  It isn't a good use of space to present the same results in both polar and Cartesian coordinates.  Also, the results from 180-360 degrees are just a mirror image of 0 to 180 degrees, so why present them at all?  Since the authors use this for all of their figures, this is an important point.  The figures are small, and unnecessarily so because there is no need to present the redundant information.  I don't think the authors ever comment on this symmetry.

357.  It's poor notation to have a two-letter variable, np.  Please use subscripts or superscripts instead if no single letter is suitable.

393.  A frequency in the THz is presented, but no size.  Most results are in dimensionless variables, which is fine.  The authors have an opportunity here to connect to the real world through the length scale and what kind of particles they want to manipulate.

Fig. 16.  The authors are using j for the imaginary unit here, but using i for most of the paper.

473.  Using

Reviewer 3 Report

Comments and Suggestions for Authors

Please find the attached review comments for this article.

Round 2

Reviewer 2 Report

Comments and Suggestions for Authors

I do not recommend publication because I do not find the results plausible.  There appears to be too sharp a definition in angle for the angular momenta, particle size, and wavelength.  Also, there are some errors in the presentation which make it difficult to know what exactly the figures mean.

Some questions from the first round review were not answered, such as why scattering is appropriate for the journal "Micromachines".  In general, the presentation is better than in the first submission.

I find Fig. 2 very difficult to believe.  There is an error in specifying ka=lambda, but suppose the authors intended ka=1.  It is difficult to understand where the strong angular dependence comes from.  On the other hand, from the drawing, it appears that a=1 um, so 2 pi a / lambda = 9.9 which might allow for the scattering distribution but differs from the authors description "a small particle relative to the wavelength".  The circle has to be the particle since it is centered at the origin which is where the particle is located.

In the case of photoemission, Ritchie Phys Rev A 13, 1411 (1976) gives a formula for low a which has a P_0(cos theta), P_1(cos theta), and P_2(cos theta) dependence for molecular photoemission.  I assume there is some closely related formula for the small a limit for scattering, probably in the references supplied by the author.

The authors should develop or present with citation the small particle limit. There should be a plots of the angular dependence of the incident beams. Perhaps Fig. 1 is trying to do this, but there is no reference to l in this plot.  The authors should demonstrate that their solutions recover analytic results in the small particle limit.  The authors should expand their discussion to include incident plane waves if there are no analytic results for Bessel beams.

The "self-healing" property of Bessel beams is not discussed in this work (except once in the introduction).  This should be a property of the scattering, or at least discussed, but it is not.

Line-by-line comments.

In Eqs. (2), q does not appear.  So the notation S_x(p,q) should be replaced by, e.g., S_x(p).  Other points which are not clear are what the meaning of u is in Eq. (1), why k appears to be in bold face in the exponential, and which of +- is for LCP.  Also, S_x(p,q) depends on l, which is not reflected in the notation.  It does not depend on x, which is in the notation.

166-168.  It's OK for the authors to study a local relation, but it should be recognized that such a relation must be nondispersive.

Eq. (3) ... why not cancel one power of k?

It would be better to use (1/c) than \sqrt(\mu_0 \epsilon_0) here and in Eq. (25).

236 and elsewhere. Units should not be in italics.

Fig. 2 caption appears to be incorrect.  Usually, k = 2 pi / lambda has the dimensions of inverse length and a is a radius with dimensions of length.  So ka is dimensionless.  Yet, the authors give ka as a multiple of lambda, which has dimensions of length.  This problem is repeated on line 248, but not 264-266.

Fig. 3 caption.  Shouldn't the (a,b), (c,f), and (d,e) be identical by symmetry?  Also, (c,d) and (e,f) are mirror images.  The pictures show this, but there is no comment.  Similarly in Figs. 4-5 (a,b), (c,d), and (e,f) are symmetry related.

Round 3

Reviewer 2 Report

Comments and Suggestions for Authors

My intuition says that the plots have too rapid angular dependence at a=lambda.   I suggest adding a couple of panels comparing the author's results to analytic results in the Rayleigh regime as well as some transition radius to see how the angular dependence evolves.    The authors can make space by reducing the number of symmetry-related figures.  It's OK to publish a pair or two of these to illustrate the symmetry, but there is no value in having many symmetry-related plots.   The extra values of a can be added to Fig. 3.   Subpanels in all of Figs. 3-9 can be removed due to the symmetry.
